# Different Approaches for the Profiling of Cancer Pathway-Related Genes in Glioblastoma Cells

**DOI:** 10.3390/ijms231810883

**Published:** 2022-09-17

**Authors:** Zuzana Majercikova, Katarina Dibdiakova, Michal Gala, Denis Horvath, Radovan Murin, Gabriel Zoldak, Jozef Hatok

**Affiliations:** 1Department of Medical Biochemistry, Jessenius Faculty of Medicine in Martin, Comenius University in Bratislava, Mala Hora 11161/4D, SK-03601 Martin, Slovakia; 2Department of Biophysics, Faculty of Science, University of P.J. Safarik, Jesenna 5, SK-04001 Kosice, Slovakia; 3Center for Interdisciplinary Biosciences, Technology and Innovation Park, University of P.J. Safarik, Trieda SNP 1, SK-04011 Kosice, Slovakia

**Keywords:** glioblastoma, cancer pathway, mRNA, multicriterial analysis

## Abstract

Deregulation of signalling pathways that regulate cell growth, survival, metabolism, and migration can frequently lead to the progression of cancer. Brain tumours are a large group of malignancies characterised by inter- and intratumoral heterogeneity, with glioblastoma (GBM) being the most aggressive and fatal. The present study aimed to characterise the expression of cancer pathway-related genes (*n* = 84) in glial tumour cell lines (A172, SW1088, and T98G). The transcriptomic data obtained by the qRT-PCR method were compared to different control groups, and the most appropriate control for subsequent interpretation of the obtained results was chosen. We analysed three widely used control groups (non-glioma cells) in glioblastoma research: Human Dermal Fibroblasts (HDFa), Normal Human Astrocytes (NHA), and commercially available mRNAs extracted from healthy human brain tissues (hRNA). The gene expression profiles of individual glioblastoma cell lines may vary due to the selection of a different control group to correlate with. Moreover, we present the original multicriterial decision making (MCDM) for the possible characterization of gene expression profiles. We observed deregulation of 75 genes out of 78 tested in the A172 cell line, while T98G and SW1088 cells exhibited changes in 72 genes. By comparing the delta cycle threshold value of the tumour groups to the mean value of the three controls, only changes in the expression of 26 genes belonging to the following pathways were identified: angiogenesis *FGF2*; apoptosis *APAF1*, *CFLAR*, *XIAP*; cellular senescence *BM1*, *ETS2*, *IGFBP5*, *IGFBP7*, *SOD1*, *TBX2*; DNA damage and repair *ERCC5*, *PPP1R15A*; epithelial to mesenchymal transition *SNAI3*, *SOX10*; hypoxia *ADM*, *ARNT*, *LDHA*; metabolism *ATP5A1*, *COX5A*, *CPT2*, *PFKL*, *UQCRFS1*; telomeres and telomerase *PINX1*, *TINF2*, *TNKS*, and *TNKS2*. We identified a human astrocyte cell line and normal human brain tissue as the appropriate control group for an in vitro model, despite the small sample size. A different method of assessing gene expression levels produced the same disparities, highlighting the need for caution when interpreting the accuracy of tumorigenesis markers.

## 1. Introduction

Glioblastoma (GBM) is one of the most prevalent primary malignant brain tumours in adults, as classified by the World Health Organization (WHO). According to histopathological and molecular characteristics, glioblastoma is classified into four grades (I-IV) [1]. Despite aggressive multimodal therapy consisting of surgical resection, radiation, and chemotherapy with the alkylating agent temozolomide, the prognosis remains dismal, with a median overall survival of 12–15 months after diagnosis [2,3]. The intensive study of malignant gliomas over the past three decades has determined several molecular hallmarks that have enhanced classification and therapeutical strategies. As with all other types of brain tumours, glioblastomas have a heterogeneous character. The result was the introduction of the most recent WHO classification of central nervous system tumours in 2021 [4,5]. In addition to the histological approach and somatic mutations, gene expression signatures contribute significantly to the overall classification of glioblastoma subtypes [6]. Currently, successful treatment response and outcome predictions for patients with GBM are made using next generation sequencing analyses that account for intratumoral heterogeneity [7]. The determination of *O*^6^-*methylguanine-DNA methyltransferase* (MGMT), *isocitrate dehydrogenase* (IDH), *tumour protein p53* (TP53), *platelet-derived growth factor receptor alpha* (PDGFRA), or *epidermal growth factor receptor* (EGFR) gene expression levels, as well as the presence of their mutations and methylation status, are important prognostic factors [8,9,10,11]. One of the most commonly used methods in the aforementioned molecular analysis is real-time PCR, indicating a clear application with minimal errors. The all real-time PCR techniques (classical method based on dyes, TaqMan probes assay, microarray, RNA sequencing analysis, etc.) are still highly quantitative and sensitive methods for the detection of gene expression levels but are generally best for examining a relatively small number of transcripts in a large set of samples. One of the few disadvantages is the necessity of involving a biostatistics expert in the evaluation process. However, with proper optimisation, we may encounter erroneous analysis and, subsequently, variable interpretations of results. Properly chosen biostatistical methods also clearly contribute to successful analysis. Cell models have also been used to characterise the mechanisms underlying glioblastoma formation [12].

In the present study, we examined the transcriptomic profiles of glial cancer cell lines: A172, T98G, and SW1088. We focused on the detection of representative cancer genes (Human Cancer PathwayFinder^TM^ PCR Array), which were divided into nine pathways: angiogenesis, apoptosis, cell cycle, cellular senescence, DNA damage and repair, epithelial-to-mesenchymal transition, hypoxia signalling, metabolism, and telomeres and telomerase (Table 1). This study’s primary objective is based on a different approach for selecting the control group and, consequently, for interpreting the results. We analysed three different control groups (non-glioma cells) widely used in glioblastoma research: Human Dermal Fibroblasts (HDFa), Normal Human Astrocytes (NHA), and commercially available mRNAs acquired from healthy human brain tissues (hRNA). Using real-time PCR analysis, the gene expression profiles of glioblastoma cell lines differ when compared to a control group based on correlational differences. Our comprehensive approach, which incorporates numerous statistical analyses, contributes to the most accurate interpretation of the results.

## 2. Results

We achieved the results on human RNA isolated from cell lines (tumour and non-tumour) and normal brain tissues. The expression of cancer pathway-related genes at the mRNA level was compared among tumour and control group of samples by the real-time qPCR. Out of the total number of monitored genes (*n* = 84) in the Human Cancer PathwayFinder^TM^ PCR Array, we were able to detect all of them, and only in the control group was there not a gene for the *Fas ligand* in any of the samples. In the next sections, we describe the relationship between the groups of samples for individual genes as well as the unique multicriteria analysis used for the correct interpretation of the results.

### 2.1. Relative mRNA Ratio of Control Cells

The selection of an appropriate control sample is a crucial initial step for group qRT-PCR analyses. On three separate controls (NHA, hRNA, and HDFa) relative mRNA gene levels associated with cancer development were detected. The Normal Human Astrocyte cell line was characterised by a significantly higher Ct ratio in the following genes: *CDH2*, *MKI67*, *LDH2*, *IGFBP7*, and *IGFBP5* (Figure 1). The *Keratin 14* gene was amplified in the NHA line exclusively. Only *Angiopoietin 1* exhibited a significantly lower mRNA ratio in NHA compared to both control groups (hRNA and HDFa). We demonstrated that there is no amplification of *AURKA*, *ANGPT2*, *FASLG*, or *GSC* products (Figure 1). *Goosecoid homeobox* gene amplification was only detected in Human Dermal Fibroblasts among the control samples. Compared to NHA and hRNA samples, the HDFa cell line exhibited overexpression of a single gene, namely *Adrenomedullin*. In contrast to the other controls, most genes were found to have lower levels. As undetected mRNA, *AURKA*, *FASLG*, *KRT14*, and *SNAI3* were considered (Figure 1a).

The total human RNA (hRNA) control sample consists of total RNAs from healthy brain donors that are commercially available. We found only hRNA control to contain the *AURKA* gene. *ANGPT2*, *CASP9*, *E2F4*, *ETS2*, *FLT1*, *GPD2*, *LPL*, *MAP2K1*, *SERPINF1*, *SNAI3*, *SOD1*, *SOX10*, *STMN1*, *TEP1*, *TERF1*, *TERF2IP*, and *TINF2* exhibited significant overexpression. In contrast, mRNA levels for *IGFBP3*, *SNAI2*, and *WEE1* were significantly higher compared to NHA and HDFa. In addition, the amplification of mRNA for Desmoplakin, Fas ligand, Goosecoid homeobox, Keratin 14, and vascular endothelial growth factor C was not detectable (Figure 1a).

### 2.2. Relative mRNA Ratio in Tumour Cell Lines

Results of the relative mRNA expression ratio comparison between A172, SW1088, and T98G cell lines are shown in Figure 1b. Similar cancer-related mRNA ratios were found in all tumour cell lines for 60 out of 84 genes (71.4%). The genes *CDH2*, *DDB2*, *DSP*, *EPO*, *FGF2*, and *TEK* were overexpressed in an astrocytoma cell line. In addition, the expression of *Cyclin D2*, *kinase insert domain receptor*, and *Keratin 14* genes was present exclusively in the A172 line. In contrast, we found a significantly lower mRNA ratio for *ACSL4*, *ADM*, *ANGPT1*, and *SNAI2*. Only the mRNA for the *Fas ligand* was not detected. Except for *CCND2*, *FLT1*, *FOXC2*, *KDR*, *KRT14*, and *TEK*, the mRNA expression ratio of cancer pathway-associated genes in the astrocytoma grade III cell line (SW1088) was comparable to that of other tumour cell lines (Figure 1b).

In the glioblastoma cell line (T98G), we detected the overexpression of the Cyclin D3 transcript. Significantly lower mRNA levels were detected for *ANGPT2*, *DSP*, *KDR*, *KRT14*, and *LPL*. Four genes (*CCND2*, *DSP*, *FASLG*, and *PGF*) lacked fluorescent signals.

### 2.3. Data Preparation before Determination of Gene Expression Level

The selection of the appropriate control is crucial when comparing mRNA levels between groups using the ΔCt method and qRT-PCR analysis. Therefore, with respect to the control group, the genes of the tested group (tumour) sought may appear to be inaccurate. Here, we used the ΔCt method by comparing the relative expression of tumour genes to that of three control genes separately to identify the validity of the results. We used a combination of several mathematical and statistical methods to prepare gene expression data.

#### 2.3.1. Principal Component Analysis

Principal component analysis (PCA) was utilised to examine the variability of gene expression profiles among various samples (see Material and Methods). Figure 2 shows the first two components, PC1 vs. PC2 as a scatter plot, with a variance of 41.2% for PC1 and 25.8% for PC2, respectively. As seen in the scatter plot, the gene profiles corresponding to tumour tissues (red circles) are closer together than the gene profiles of controls. From an overall perspective, tumour lines can be defined by a putative cluster, whereas other data are more dispersed. Expression profiles of controls demonstrate a wide variation, indicating their distinct and cell-line-specific expression. Regarding variability within the same sample–experimental replicas, only small variations are visible, and all replicas can be attributed to a given cell line.

#### 2.3.2. Correlations between Gene Expressions

In the previous analysis, we discovered that LDA can identify and pinpoint the differences in individual genes and distinguish between the classes of control cell lines and tumour samples. Subsequently, we have decided to analyse the relationships between gene expressions in greater detail. By calculating the Pearson correlation coefficient, we focused on the relationship between the expressions of individual genes. In control cells, a general gene regulation network controls gene expression. Therefore, significant positive and negative correlations between gene expressions will result from the regulation network. Gene correlations can be lost if the regulatory network is dysregulated, such as in tumour cells. Figure 3a depicts a correlation between the genes in the control samples and the genes in the tumour samples. In control samples, the average correlation is low (0.33), but there are substantial differences between the individual genes. For example, *TERF1* and *TERF2IPa* genes had the highest correlation of 0.999, and *SNAI2* and *CASP9* genes had the highest anti-correlation of −0.996. In tumour lines, the average correlation is even lower, 0.031, indicating a significant loss of coordinated gene expressions. Given that the overall expression profiles of individual genes in tumour cells are more similar (dots in red elliptical line, Figure 2) than in control cells, which appeared more heterogeneously in the PCA plot, this may be a surprising result. The difference between PCA and gene–gene correlation is that PCA describes the variance between full profiles, whereas gene–gene correlation provides information about possible causal relations between expression values. In tumour cell lines, the relationship between the *SNAI1* and *SNAI2* genes is the highest at 0.9999, while the *LPL* and *IGFBP5* genes have the highest anti-correlation (−0.997). The descending arrangement of the gene correlation (Figure 3b) indicates that tumour samples have lower overall values than control samples.

Kernel density estimation revealed a different relationship between gene expressions (Figure 4). The distribution of gene expression correlations deviates from the Gaussian distribution for both control and tumour cell lines. In contrast to tumours, the peak density of control samples is centred around the value of 0.95. Maximum density in the positive correlation range is roughly in the same place as in control cell lines, but the amplitude is slightly lower, indicating that some gene–gene correlations have changed. The missing positive gene–gene correlations have become negative in tumour cell lines. The density analysis of the control vs. tumour correlation values approaches 0, as shown in Figure 4.

In the next step, we calculated Pearson correlations between gene expression pro-files of analysed cell lines (Figure 5). First, the correlation of gene expression profiles within the control group and within the tumour group was calculated. We found that the average Pearson correlation coefficient among the control group is 0.71, which is slightly lower than the average correlation between tumour cells at 0.84. When we pairwise correlated the expression profiles of control and tumour cells, we arrived at the average correlation coefficient of 0.56. Some of the gene expression profiles of control vs. tumour cells showed higher correlation coefficients than profiles of controls-only or tumour-only. For example, the profiles of HDFa_3 versus the expression profile of hRNA_1 control shows a correlation coefficient of 0.47. Conversely, the expression profile of HDFa_3 versus expression profile of SW1088_3 from the tumour group shows the correlation coefficient of 0.72.

### 2.4. Fold Change of Cancer Pathway-Related Genes

Figure 6 shows the fold regulation data obtained by correlating cancer cell lines (A172, SW1088, and T98G) with three distinct control groups (HDFa, NHA, and hRNA). Fold downregulation is represented by blue values, while fold upregulation is represented by red values. Changes in regulation that do not reach significance are highlighted in grey. Not available values (N/A) representing changes in expression are marked in white.

When comparing the expression of genes associated with angiogenesis, we obtained heterogeneous results (Figure 6). The significantly higher expression of *ANGPT1* was detected in the T98G glioblastoma cell line (2.20 compared to HDFa; 2.48 compared to hRNA; 27.98 compared to NHA; and 5.48 compared to all controls) and the SW1088 cell line compared to the NHA control (16.96). In contrast, data analysis of Angiopoietin 1 in A172 revealed a statistically significant decrease in expression levels relative to HDFa and hRNA controls. At the same time, the expression of the Angiopoietin 1 antagonist coding gene, *ANGPT2*, was found to be opposite in comparisons between SW1088 and a combination of controls (5.35). The fold regulation of *ANGPT2* was significantly lower in all tested cell lines compared to control RNAs isolated from the whole brain. In contrast, A172 and SW1088 lines exhibited elevated expression levels (7.57 and 56.2) compared to HDFa control. Compared to control samples, the expression of *CCL2* appeared to be stably increased across tested cell lines, whereas the expression of *FGF2* appeared to be stably decreased. We discovered the reduced expression of *FLT1* and *PGF*. *FLT1* expression was statistically significantly decreased in both glioblastoma cell lines, whereas *PGF* was not detected in glioblastoma cell line T98G (Figure 6).

Among the investigated apoptosis genes, lower expression of *APAF1*, *BIRC3*, *CFLAR*, and *XIAP* was detected in all test groups, with statistical significance and fold regulation varying according to divergent control groups (Figure 6). For instance, in the A172 group, expression of *BIRC3* was three times lower in correlation with NHA than in the hRNA control group. The same pattern was observed when fold regulation values were correlated with HDFa and hRNA. In the astrocytoma cell line, we found that the gene encoding Caspase 2 was upregulated in relation to HDFa control (3.15). *CASP2* deregulation was not significant in either glioblastoma cell line, but it was constantly increasing. In contrast to *CASP2*, the expression of another caspase family protease, *CASP9*, was inconsistently deregulated (Figure 6). CASP9 expression was only found to be negatively regulated in glioma cancer cell lines when compared to the hRNA control group. However, we found that all lines correlated to NHA had relatively high positive fold changes (Figure 6).

Our results in cell cycle-related genes indicate, with a few exceptions, a significant increase in the expression of genes involved in cell division (*CDC20*, *E2F4*, *MCM2*, *MKI67*, and *SKP2*). Notable is the similarity between the expression fold changes of the *CDC20* and *E2F4* genes in the control group (Figure 6). Decreased expression of *CCND3* was prominently identified in SW1088 in terms of hRNA and NHA control groups (−7.52 and −2.45). The expression of *Stathmin* was found to be decreased in all test groups. However, there are differences in fold regulation relative to controls. *WEE1* expression was found to be statistically significant in all glioma cell lines, but only when compared to hRNA controls (4.75, 2.79, 2.24).

We observed decreased expression of *BMI1*, *ETS2*, *IGFBP5*, *IGFBP7*, *SOD1*, and *TBX2* from all investigated genes associated with cellular ageing molecular pathways, but differences in negative fold regulation with respect to control groups were remarkable (Figure 6). For instance, among all glioma cancer cell lines, the fold change of *IGFBP7* expression was decreased approximately 16.5 times more in comparison to the NHA control group than in comparison to the hRNA control and approximately 2.5 times more in comparison to human dermal fibroblasts. To mention the regulation of mitogen-activated kinase’s expression in glioblastoma cell lines, *MAP2K1*, *MAP2K3*, and *MAPK14* genes were only upregulated when compared to HDFa and NHA. All three MAP kinases within the tested cell lines were downregulated with varying statistical significance in terms of hRNA control.

Among the eight investigated genes involved in the response to DNA damage in the A172 group, *DDB2*, *GADD45G*, and *POLB* were significantly upregulated when compared to the HDFa control group (Figure 6). There was downregulation of *LIG4* in glioma cancer cell lines. In A172, the expression of this gene was approximately 4.2 times lower when compared to *LIG4* expression in hRNA control. SW1088 and T98G cell lines exhibit a similar pattern. Despite differences in fold change values due to control group selection, other statistically significant results from Figure 6 were broadly in line with expectations. *SNAI1* and *SNAI2* were downregulated genes related to the EMT pathway that we discovered in all cell lines, but only with regard to NHA control (Figure 6). When compared to HDFa and hRNA control groups, the expression of *SNAI1* and *SNAI2* in SW1088 and T98G lines was increased. In terms of hRNA, all glioma cell lines showed a significant decrease in *SOX10* expression (−4502; −26,634; −19,800), indicating that *SOX10* is highly expressed in our normal human brain control. In comparison to HDFa and NHA controls, *SOX10* appeared to be downregulated. Other detected downregulated genes were *CDH2*, *FOXC2* and *OCLN*. Downregulation of *FOXC2* was present only in the glioblastoma cell line. Fold regulation in SW1088 cell lines was not determined.

Analysis of hypoxia-signalling factors revealed a decrease in expression in the majority of genes, including *SLC2A1*, the gene- encoding Glucose transporter type 1 (Figure 6). This result is statistically significant only in SW1088 (−7.31; −32.3; −12.8) when compared to the hRNA control in the A172 cell line (−3.89). The Erythropoietin-encoding gene *EPO* appears to be exclusively downregulated in SW1088. In contrast, fold change was detected without statistical significance in the other two cell lines (A172) or was not detected at all (T98G).

Most cellular metabolism genes investigated in glioma cell lines (*ATP5A1*, *COX5A*, *CPT2*, *PFKL*, and *UQCRFS1*) were found to be statistically downregulated. *ACSL4* was found to be downregulated in only one of the glioma cell lines, A172. Although *GPD2* and *LPL* were found to be upregulated in A172 and SW1088, with similar fold changes and statistical significance when compared to HDFa, NHA, and a combination of controls, we discovered that both genes were downregulated when compared to the hRNA control group. In contrast, *LPL* expression in T98G was significantly reduced, with the lowest achieved value in comparison to the hRNA control group. *G6PD* upregulation was revealed to be statistically significant only in the SW1088 (5.46; 2.67; 3.34) and T98G (13.7; 6.67; 8.35) cell lines (Figure 6). We also observed increased expression of *ACLY*, but none of these differences was statistically significant.

Of the eight investigated genes involved in telomere maintenance and telomerase regulation, three (*PINX1*, *TNKS*, and *TNKS2*) were found to be downregulated in all glioma cell lines when compared to individual controls (Figure 6). No significant differences in fold changes of *PINX1* expression were found in A172 or SW1088 in comparison to all three negative controls (Figure 6). The fold change in expression in T98G was approximately two times lower (−2.31; −4.74; −3.98). We identified an increase in expression of *TEP1*, *TERF1*, and *TERF2IP* in glioblastoma and astrocytoma cell lines compared to HDFa and NHA control cell lines. The study found no statistically significant changes in *TERF2IP* expression in the T98G cell line. In contrast, we found a statistically significant decrease in *TEP1*, *TERF1*, and *TERF2IP* expression when compared to hRNA control.

### 2.5. Multi-Criteria Decision of Gene Expression in Sample Correlation

To determine the power of individual genes in groups, a multi-criteria decision support system was used. As a result, we were able to identify a group of genes in the correlation between controls and tumours.

The alternative method for assessing gene expression is based on separate, two-value quantification of the consequences of over- or underexpression. Not only is knowledge of the data carriers and gene expression important here, but so is the emphasis on the form of the set ordering in sets of triplicates. The results in the plots are represented by a normalised (dimensionless) two-factor form called *R*^+^, *R*^−^. Each *R* represents a proportion of the numerator’s dissimilarity tendencies. Both measures preferentially account for when there is either an increase (superscript +) or decrease (superscript −) in gene expression in the test subject relative to the control. The denominators in the measures only serve for normalisation because they track intra-group differences in expression. For a robust linear combinatorial representation of stochastic measures, we proposed a data-driven approach inspired by the weighting theory of measures derived from multi-criteria decision support systems theory [11]. The details of our procedures for deriving *R*^+^, *R*^−^, as well as the methods of calculation, are difficult to explain concisely; therefore, we provide a detailed description of them in the Appendix A.

Figure 7 depicts the resultant correlation of genes (up or down) for individual pathways. As mentioned above, the individual points represent triplicate measurements of the logarithmic values of delta Ct for all samples, correlating the control and tumour groups. Correlation 2D plots of in-plane gene expression (*R*^+^, *R*^−^) constructed for tumour cell lines include comparison with controls. A 45-degree line separates the regions of overexpression and underexpression; at this line, overexpression and underexpression are therefore balanced, the scenarios when noise predominates are covered by the (0.1) × (0.1) square. This square region is hence statistically less important. In addition to the shown genes, it is necessary to mention points that far exceeded our proposed trend scale and were therefore deemed irrelevant. We identified the following genes as having decreased expression in tumour lines compared to controls: *FGF2*, *APAF1*, *CFLAR*, *XIAP*, *STMN1*, *WEE1*, *ERCC5*, *LIG4*, *PPP1R15A*, *OCLN*, *SOX10*, *ARNT*, *LDHA*, *ATP5A1*, *COX5A*, *CPT2*, *PFKL*, *UQCRFS1*, *ETS2*, *IGFBP3*, *IGFBP5*, *IGFBP7*, *SOD1*, *TBX2*, *PINX1*, *TINF2*, *TNKS*, and *TNKS2* (out of range). *CCL2*, *CASP2*, *CDC20* (out of range), *E2F4*, *MCM2*, *MKI67*, *SKP2*, *ACLY*, *G6PD*, *GPD2*, *MAPK14*, *DKC1*, and *TERF1* were the genes with increased levels in the tumour lines.

## 3. Discussion

Glial tumours are biologically aggressive neoplasms with an abnormally high proliferative capacity and a diffuse invasion pattern. Glioblastoma (grade IV astrocytoma), composed of poorly differentiated neoplastic astrocytes, is the most malignant astrocytic tumour. Based on histopathological and molecular criteria, the WHO grading system categorises gliomas into grades I through IV [4]. Although the majority of neurological tumours derive from the glial lineage, it is unknown whether tumour cells arise from the transformation of an immature precursor or the dedifferentiation of a mature glial cell. Several genetic pathways are involved in the initiation and progression of these neoplasms, especially in the emergence of secondary GBMs.

In our study, we focused on the transcriptomic analysis of genes associated with the cancer pathways in glial tumour cells. As the experimental models, we chose the human glioma cell lines A172 (glioblastoma), SW1088 (astrocytoma), and T98G (glioblastoma). A172 and T98G cell lines are currently the most commonly used glioma cell lines for gene expression analysis. Weller’s team performed the first large-scale analysis of 12 glioma cell lines, estimating the profile of 5800 genes. Their cluster and gene expression correlation analysis identified subsets of genes whose expression levels exhibited significant associations with drug sensitivity profiles [13]. Kiseleva et al. identified morphological, surface markers, and several growth factor genes or extracellular matrix genes in the characterisation of both glioblastomas, A172 and T98G [14]. Among the nine tested genes, the expression of *Alpha actin 2* was notably high in both cell lines. In addition, the data revealed a high level of activity of genes encoding major angiogenesis inducers (*VEGF*, *FGF2*, *TGFb1*) and *Thrombospondin-1*. The transcriptomic analysis of SW1088 cells was associated with individual genes or various inhibitory effects [15,16,17]. In our previous study, we determined the effect of ABT-737 and MIM-1 inhibitors on the mRNA level of apoptosis-associated genes in the T98G cell line [18]. As a control group, human astrocyte (HA) cells were used. Significant changes in apoptotic gene expression were obtained in both cell lines, with the greatest number of altered genes (*n* = 42) occurring in the HA line following MIM-1 treatment. Regardless of the genes involved in determining fold regulation between groups of samples, the choice of control samples will always be decisive. In addition, based on our final multivariate criterion, it is evident that results vary not only according to the choice of control but also according to the evaluation method employed. Therefore, the discussion will centre solely on the genes selected using the MCDA method.

Angiogenesis, as one of the hallmarks of cancer [19], plays a crucial role in glioblastoma growth through oncogene activation and/or downregulation of tumour suppressor genes, resulting in the upregulation of angiogenic pathways [20]. The initial step in the induction of angiogenesis in GBM is the overexpression and secretion of angiogenic growth factors, such as vascular endothelial growth factor (VEGF), followed by their binding to the receptors on epithelial cells [21]. Fibroblast growth factor 2 encoded by the *FGF2* gene is a crucial positive regulator of glioblastoma cell proliferation and survival [22]. However, our results showed a decrease in *FGF2* expression. Although the loss of the *FGF2 receptor* gene is associated with a poor prognosis in glioma patients [23], *FGF2* seems to be persistently expressed because it has been identified as an oncogenic factor in GBM [24], and its expression has been confirmed in other gliomas and meningiomas [25].

Apoptosis refers to a programmed cell death characterised by non-inflammatory cellular fragmentation [26]. It is an essential regulatory mechanism for cell proliferation and death. Intrinsic or extrinsic pathways can initiate apoptosis, with both leading to proteolytic activation of caspases and controlled cell death. Cancer cells have evolved mechanisms to sustain proliferative signalling, thereby sustaining cell growth and avoiding cell death. In the current study, only nine apoptotic genes were included in the array, with *caspase-2* and *caspase-7* being the most expressed in tumour lines. The remaining genes involved in apoptosis regulation were downregulated, including *CFLAR*, *XIAP* and *APAF1*. X-linked inhibitor of apoptosis protein is the most potent and best-defined anti-apoptotic IAP family member that directly counteracts apoptosis by binding to caspase-9 and the effector caspases-3 and -7 [27]. The X-linked inhibitor of apoptosis protein is abnormally expressed in a variety of human cancers [28,29]. Although this is not evident in brain tissue, Murphy et al. investigated the low levels of XIAP in GBM patients and brain cell lines [30]. The protein encoded by the antiapoptotic gene *CFLAR* is a Caspase-8 and FADD-like apoptosis regulator. By binding to the death receptor, it protects cells from cell death signalling and inhibits receptor-mediated apoptosis [31]. Induction of hypoxia in the A172 glioblastoma cell line results in the expression of *CFLAR* [32]. *CFLAR* expression was detected in glioblastoma tissue samples [33]. Despite these findings, our analysis showed a significant reduction in *CFLAR* expression. Apoptotic peptidase activating factor 1 (APAF1) is a proapoptotic protein that participates in the formation of apoptosomes in response to cell death signals [34]. Overexpression of *APAF1* induced apoptosis in U-373MG human glioma cells [35]. Our previous study on apoptotic gene expression revealed a slight decrease in the expression of *APAF1* in glioblastoma patient samples [36]. In this study, we confirmed our previous findings regarding glioma cell lines. A decrease in *APAF1* expression may lead to apoptosis reduction, thereby favouring cancer cell survival [36].

The main goal of the cell cycle is to ensure accurate DNA replication in the S phase and the final formation of two identical daughter cells in the mitotic phase. The cells use various checkpoints to maintain the optimal progression of the cell cycle, which will slow down or stop the event if necessary [37]. Ki-67, a prognostic and proliferative marker expressed by the *MKI67* gene in cell nuclei during the active phases of the cell cycle (G1-M) with maximum expression at the G2/M phase interface, is used to control the malignant nature of cells [38]. In gliomas, its elevated expression, which increases with malignancy grade, has been well characterised [39]. The absolute highest expression of all genes was observed in *CDC20*. The protein of the same name is responsible for regulating the mitotic phase of the cell cycle. Jeremy Rich’s team identified an increased expression of *CDC20* in GBM compared to lower grade gliomas and healthy brain tissue. Their results also indicate the importance of *CDC20* proto-oncogene expression in glioblastoma stem cells, as it plays an essential role in the regulation of proliferation, self-renewal, and survival of these cells [40]. It even contributes to glioma chemoresistance. In accordance with the aforementioned studies, we also identified upregulation of the *CDC20* gene in all monitored groups.

Stathmin is an oncoprotein “18” that is distributed throughout the cytoplasm of cells and regulates microtubule kinetics, thereby affecting cell cycle proliferation and differentiation. Many studies indicate that *STMN1* expression is elevated in glioblastomas [41] and a variety of human cancers [42,43]. Our analyses identified a significant downregulation of *STMN1* in all tumour lines relative to the average control, suggesting reduced cell proliferation and tumour cell migration. The essential cell cycle regulator *WEE1* kinase was similarly underexpressed in tumour cell lines. Its primary function is to stop the progression of the cell cycle at the transition from G2 to the mitotic phase in cells with defectively replicated or damaged DNA [44]. In addition, glioblastoma patients whose *WEE1* expression is upregulated have a shorter survival rate [45,46]. We identified elevated levels of the *WEE1* gene in all tumour lines when compared to human RNA from healthy brain tissues.

The main role of DNA repair mechanisms is to respond to environmental factors that cause DNA damage [47]. These gene mutations can result in a diminished or impaired capacity to repair DNA and an accumulation of damaged DNA, which ultimately increases the risk of cancer. Furthermore, tumour cells overexpress the genes encoding DNA repair mechanisms, increasing repair capacity and treatment resistance [48]. DNA ligase IV joins single-strand breaks in a double-stranded polydeoxynucleotide in an ATP-dependent reaction, and its low expression results in inefficient function of the repair system. Compared to normal astrocytes, brain tumour lines had lower levels of the *LIG4* coding gene, and these findings correlated with transcriptomic and genomic analyses [49]. In all tumour cell lines, decreased *LIG4* expression was observed. We also observed a significant reduction in expression of the *ERCC5* gene, whose product is part of the nucleotide excision repair system [50]. Borderline low levels of the gene encoding Protein Phosphatase 1 Regulatory Subunit 15A have been confirmed.

Epithelial to mesenchymal transition (EMT) is the process by which epithelial cells lose their epithelial characteristics and acquire a mesenchymal phenotype, resulting in increased mobility and chemoresistance [51]. Although glioma cells are not of epithelial origin, an EMT-like process in GBM can be induced [52]. Overexpression of various growth factors, such as transforming growth factor (TGF), epidermal growth factor (EGF), fibroblast growth factor (FGF), and HIF-1, cause EMT in cancer cells [53]. As a result of growth factor-mediated signalling, transcription factors (Snail, Slug, dEF1, SIP1, Twist1, and FOXC2) are activated and induce an EMT-like phenotype [54]. Multiple signalling pathways participate in these processes. The phosphatidylinositol-3-kinase (PI3K)/Akt signalling pathway plays an important role in regulating cell growth and maintaining cancer biology. Cooperation with other signalling pathways such as transforming growth factor β (TGF-β), nuclear factor (NF)-κB, and Ras and Wnt signalling pathways leads to direct or indirect induction of the EMT process, resulting in enhanced invasiveness, aggression, chemoresistance, and apoptosis resistance of the tumour mass [55]. The transcription factor *SOX10* is one of the key determinants of oligodendroglial differentiation. Therefore, Bannykh and colleagues decided to compare the presence of *SOX10* in oligodendrogliomas and astrocytomas to determine its specificity. Although at lower levels [56], the majority of oligodendrogliomas and a significant proportion of astrocytomas, including glioblastomas, produced *SOX10*. Consistent with previous research, multivariable analysis confirmed a decrease in the expression of *SOX10* and *OCLN*, which belongs to the EMT group [57,58].

Cancer cells surrounding the necrotic nucleus lack nutrients and oxygen. Hypoxia is the primary physiological trigger of angiogenesis [59], which is activated by Hypoxia-inducible factor 1 [60]. *HMOX1* is one of the many genes expressed during hypoxia induced by HIF-1. Due to its antioxidant and antiapoptotic effects, Hemoxigenase 1 plays a crucial role in tumour growth [61]. Because *HMOX1* activity stimulates angiogenesis, this enzyme is a suitable indicator of glioma neovascularization [62]. Only the T98G glioblastoma cell line was found to have elevated levels of the *HMOX1* gene. In contrast, only *LDHA* and *ARNT* were downregulated relative to controls. Lactate dehydrogenase A (LDHA) is a key enzyme in the anaerobic glycolytic pathway [63]. In addition to promoting acidification of the microenvironment, lactate production promotes the metastatic nature of the tumour [64]. Several authors have reported on the significance of glioblastoma *LDHA* expression [65,66,67]. Chesnelong and colleagues found low expression and high methylation of *LDHA* in IDHmt glioblastomas [68]. Kathagen-Buhmann et al. identified a decline in LDHA production in non-migrated cells [67]. Since *LDHA* expression is promoted by hypoxia, low levels of *LDHA* in gliomas that were cultivated under standard conditions in the presence of oxygen may be attributable to an oxygenated environment.

The largest number of changes in gene expression (*n* = 8) between tumour and non-malignant groups were identified in metabolic genes (*PFKL*, *ATP5A1*, *UQCRFS1*, *CPT2*, *COX5A*, *ACLY*, *GPD2*, and *G6PD*). Even under aerobic conditions, tumour cells are known for their high glycolytic activity [69]. Along with an increase in glucose consumption and lactate production, this promotes rapid cell proliferation and GBM growth, which is correlated with the elevated activity of glycolytic enzymes [70]. Phosphofructokinase-1 is a regulatory glycolytic enzyme catalysing the phosphorylation of fructose-6-phosphate to fructose-1,6-bisphosphate. The presence of its liver isoform (PFKL) in gliomas was analysed by Stanke et al. However, they did not observe any statistically significant changes compared to healthy tissue [71]. In our samples, we identified a statistically significant decrease in *PFKL* expression, confirming that the prevalent isoform in brain tissue is the platelet isoform (PFKP), not PFKL [72]. A high expression of *Glucose-6-phosphate dehydrogenase* was spotted. Glucose-6-phosphate dehydrogenase (G6PD) is one of the pentose phosphate pathway (PPP) enzymes that catalyses the production of NADPH [73]. During normoxia, glioma and non-neoplastic brain cells both produce an abundance of these enzymes. A negative association between *G6PD* expression and survival in patients with low-grade glioma was discovered [74]. COX5A is one of the three subunits of cytochrome c oxidase, a respiratory chain complex IV encoded by mitochondrial DNA [75]. UQCRFS1 is a respiratory chain complex III subunit. Both *COX5A* and the gene encoding another subunit of complex III, *UQCRB*, are downregulated in glioblastoma patients compared to healthy individuals. In contrast, when compared to expression in gliomas of lower grade malignancy, *COX5A* expression is significantly increased in GBM, and *UQCRB* expression is at approximately the same level. The reduced expression has also been linked to a poor prognosis [71]. In line with previous findings, we were able to identify the downregulation of *COX5A* and *UQCRFS1*.

Cellular senescence is an irreversible process of cell cycle arrest [76]. During this process, senescent cells undergo morphological changes that include flattening, increased cytoplasmic volume, or increased granularity. Only the *MAPK14* gene was found to be overexpressed in our sample cohort, while six other genes were found to be underexpressed (*IGFBP3*, *IGFBP5*, *IGFBP7*, *SOD1*, *TBX2*, and *ETS2*). *ETS2* is a transcription factor that regulates apoptotic and angiogenic genes, as well as genes involved in proliferation and differentiation [77]. Cam et al. identified *ETS2* expression in glioblastomas and, in association with ΔNp73, confirmed its role in tumour progression, angiogenesis, and improved tumour cell survival [78]. On the contrary, bioinformatic analysis of transcriptomic data from glioma patients revealed a decrease in gene expression of *ETS2* regardless of the degree of malignancy [79]. In glioma cell lines, we found a statistically significant decrease in *ETS2* expression. Superoxide dismutase 1, encoded by the *SOD1* gene, is an enzyme that converts free superoxide radicals into less harmful hydrogen peroxide and oxygen [80]. A decrease in *SOD1* expression has been identified in glioblastomas and is associated with improved response to radiotherapy and a better prognosis for patients [81,82]. *SOD1* expression was also reduced in glioma cell lines when compared to non-malignant cell controls. The only overexpressed gene involved in the regulation of senescence is *MAPK14*. Mitogen-activated protein kinase 14, a protein product of *MAPK14*, is an essential component of the MAP kinase signal transduction pathway that influences the direct activation of transcription factors in response to cell stress stimuli [83]. *MAPK14* expression was found to be elevated in glioma cells, which is in contrast to the findings of other studies, which indicated that the expression of this gene was decreased in glioblastoma samples [84].

Human telomeres, located at the ends of chromatids, are tandem nucleotide repeats of a short DNA sequence associated with various telomere-binding proteins with a predominantly protective function [85]. The primary function of telomeres is to compensate for incomplete DNA replication at chromosome ends, thereby maintaining intact genetic information [86]. However, as a result of cell division, telomeres become progressively shorter, resulting in cellular senescence and apoptosis induction [87]. PINX1 was identified as a potent telomerase inhibitor that interacts directly with the catalytic activity of telomerase [88]. Our analysis revealed a decrease in *PINX1* expression. Previous studies have shown a correlation between a decrease in *PINX1* expression and the metastatic nature and poor prognosis of cancer patients [89]. In glioblastoma cell lines with induced overexpression of *PINX1*, there was a reduction in cell migration and proliferation due to cell cycle arrest at the G1 phase [90]. In contrast, there is evidence that *PINX1* expression is associated with poor survival in glioma patients because it promotes cell proliferation [75,91]. The *DKC1* gene encodes Dyskerin, an additional protein that regulates telomerase activity [92]. Glioma is one of several human cancers in which *DKC1* is upregulated [93,94]. Consistent with previous findings, elevated expression of *DKC1* in glioma cell lines was also identified. Tankyrases (TNKS, TNKS2) are proteins involved in telomere length maintenance [95], which, together with regulation of the Wnt/β-catenin pathway, is important for cancer cell renewal and survival [96]. Expression of *TNKS* and *TNKS2* was decreased in glioma cells compared to non-malignant cells and normal brain tissue. Additionally, the expression of *TINF2* was reduced. We identified an increase in *TERF* gene expression.

Glial tumours are biologically aggressive neoplasms with an elevated, often aberrant, and diffusely invading proliferative capacity. Composed of poorly differentiated neoplastic astrocytes, glioblastoma (grade IV astrocytoma) is the most malignant astrocytic tumour. According to histopathological and molecular criteria, the WHO grading system categorises gliomas into grades I through IV, based on their degree of malignancy. Although the majority of neurological tumours derive from the glial lineage, it is unclear whether tumour cells result from the transformation of an immature precursor or the dedifferentiation of a mature glial cell. Several genetic pathways are involved in the initiation and progression of these neoplasms, particularly during the manifestation of secondary GBMs.

## 4. Materials and Methods

### 4.1. Cell Culturing

The glioma tumour cell panel (T98G, A172 and SW1088) was purchased from American Type Culture Collection (ATCC) under catalogue numbers: CRL-1690™, CRL-1620™, HTB-12™; respectively). Cell cultures were maintained as monolayer in Dulbecco’s modified Eagle’s media with 25 mM glucose, and supplemented with foetal bovine serum (10%, *v*/*v*), and penicillin/streptomycin (1×; PAA). Normal Human Astrocytes (NHA) were provided from ATCC and cultured in Dulbecco’s modified Eagle’s media high glucose/F12 (1:1; Merck KGaA, Darmstadt, Germany) supplemented with foetal bovine serum (10%, *v*/*v*), and penicillin/streptomycin (1×; PAA). The Human Dermal Fibroblasts (HDFa; Gibco—Thermo Fisher Scientific, Waltham, MA, USA) was used as non-specific tissue control and cultured in HAM´s Nutrient Mixture F12 (Merck KGaA, Darmstadt, Germany) supplemented with foetal bovine serum (10% *v*/*v*; Gibco—Thermo Fisher Scientific, Waltham, MA, USA) and penicillin/streptomycin (1×; PAA Laboratories GmbH, Austria). Cells were cultured at 37 °C in an atmosphere of 5% CO_2_. Before each experiment, single-cell suspension was prepared using 0.05% trypsin/EDTA solution, and cells were counted using Countess^TM^ automated cell counter (Thermo Fisher Scientific, Waltham, MA, USA).

### 4.2. Control Brain RNA

Commercially available total RNA from human brain tissue of single healthy normal donor was used as a control group (HR-201, Human Brain Total RNA—Amsbio, Abingdon, UK) for quantitative PCR. For quantitative PCR analysis, we used three independent transcripts into cDNA.

### 4.3. RNA Extraction and cDNA Synthesis

Total RNA was isolated using AllPrep^®^ DNA/RNA Mini Kit (Qiagen Inc., Germantown, MD, USA). Concentration of isolated RNA was measured in Implen P300 NanoPhotometer (Implen GmbH, München, Germany). Two micrograms of purified cellular RNA was converted to single-stranded cDNA using RT^2^ First Strand Kit (330,401; Qiagen Inc., Germantown, MD, USA) according to the protocol supplied by the manufacturer.

### 4.4. Real-Time PCR Array

Real-time PCR (quantitative PCR) was carried out using RT^2^ SYBR^®^ Green Rox^TM^ qPCR Mastermix (330,502; Qiagen Inc., Germantown, MD, USA) in 96-well plate format of the Human Cancer PathwayFinder^TM^ PCR Array (PAHS-033ZC; Qiagen Inc., USA). The PCR reaction mix (SYBR^®^ Green Rox^TM^ qPCR Mastermix (1340 μL), PCR water (1290 μL) and cDNA (50 μL) was distributed into the 96-well plate to a final volume of 25 μL per well. The sealed plate was briefly centrifuged at 1000× *g* for 1 min. Amplification was performed in the ViiA7 Real-Time PCR system (Thermo Fisher Scientific, Waltham, MA, USA). After denaturation at 95 °C for 10 min, fluorescence was detected over 40 cycles (95 °C for 15 s, 60 °C for 1 min).

### 4.5. Statistical Analysis

Samples of cDNA were measured in triplicate, and the levels of the genes of interest were normalized to the three endogenous controls (β-actin, *ACTB*; Ribosomal protein large unit P0, *RPLP0* and Glyceraldehyde-3-phosphate dehydrogenase, *GAPDH*), determined using the ΔΔCt method. The expression data from the separate control group:Human Dermal Fibroblasts (HDFa);Normal Human Astrocytes (NHA);Human Brain Total RNA (hRNA).

These were used as a reference in the ΔΔCt method calculation for each glial cell line (T98G, A172 and SW1088) individually. The relative expression of 84 genes in tumour cell lines and non-neoplastic samples was calculated using the RT^2^ Profiler PCR Array Data Analysis Web Portal (Qiagen) based on 2^−ΔΔCt^ method [97], where ΔΔCt = (Ct_GOI_ − Ct_HKG_)_TESTING GROUP_ − (Ct_GOI_ − Ct_HKG_)_CONTROL GROUP_. Fold-change calculations were performed using Qiagen data analysis software (https://dataanalysis2.qiagen.com/pcr, accessed on 1 January 2022). The genes with a significant difference in expression were those with an average fold-change of ≤−2.0 or ≥2.0, and statistically significant differences were those with a corresponding *p* value of <0.05.

Gene expression values were normalised to a 0–1 scale for both control and tumour cell lines. Using the KNIME Analytics Tool, sample normalisation, principal component analysis, and linear discriminant analysis were calculated.

For the statistical analyses mentioned above, only genes with detectable signals in all samples were selected from the raw data set. The Euclidean distance was used to calculate the distance between the samples. Using Pearson correlation, gene–gene expressions of control and tumour samples were correlated, respectively. The Pearson correlation coefficient for the two populations (*X*, and *Y*) is calculated as follows:$$\rho_{X,Y}=\frac{cov\left(X,Y\right)}{\sigma_{X}\sigma_{Y}}$$
where *cov*(*X*,*Y*) is the covariance; *σ_X_* is the standard deviation of *X*; and *σ_Y_* is the standard deviation of *Y*. The analyses were calculated in Python using Anaconda Navigator and JupyterLab. The Pandas, NumPy, and SciPy libraries were used. Kernel density and bandwidth optimisation were calculated using the Shimazaki and Shinomoto web application (https://www.neuralengine.org/res/kernel.html, accessed on 8 March 2022). We used the Matplotlib Python library to visualise the heatmap of gene correlations.

## 5. Conclusions

We focused on the transcriptomic analysis of genes associated with cancer pathways in glial tumour cells. As the experimental models, we selected the human glioblastoma cell lines A172 and T98G and the astrocytoma cell line SW1088. Sixty genes were deregulated in glioblastoma cell line A172 in comparison to the HDFa control group; 57 genes in comparison to the human RNA control group; and 54 genes in comparison to the human astrocytes control group, according to transcriptomic data. In the astrocytoma cell line SW1088, we found differences in the expression levels of 57, 60, and 59 genes related to HDFa, hRNA, and NHA control groups, respectively. In correlation with T98G and HDFa, 47 significantly deregulated genes were discovered. With hRNA, 57 genes, and the NHA control group, 52 genes with varying expression levels were identified. By combining the PCA method and multi-criteria decision in the analysis of gene expression, we were able to identify altered genes involved in cancer pathways in heterogeneous sample groups. We managed to reduce the selection of significant genes based on a combined mathematical analysis. In tumour cells, we finally identified 26 genes that showed a deregulated state compared to the average expression value of three different controls. The most changed genes represented pathways involved in cellular senescence (*BM1*, *ETS2*, *IGFBP5*, *IGFBP7*, *SOD1* and *TBX2*) and then metabolism (*ATP5A1*, *COX5A*, *CPT2*, *PFKL*, *UQCRFS1*).

## Figures and Tables

**Figure 1 ijms-23-10883-f001:**
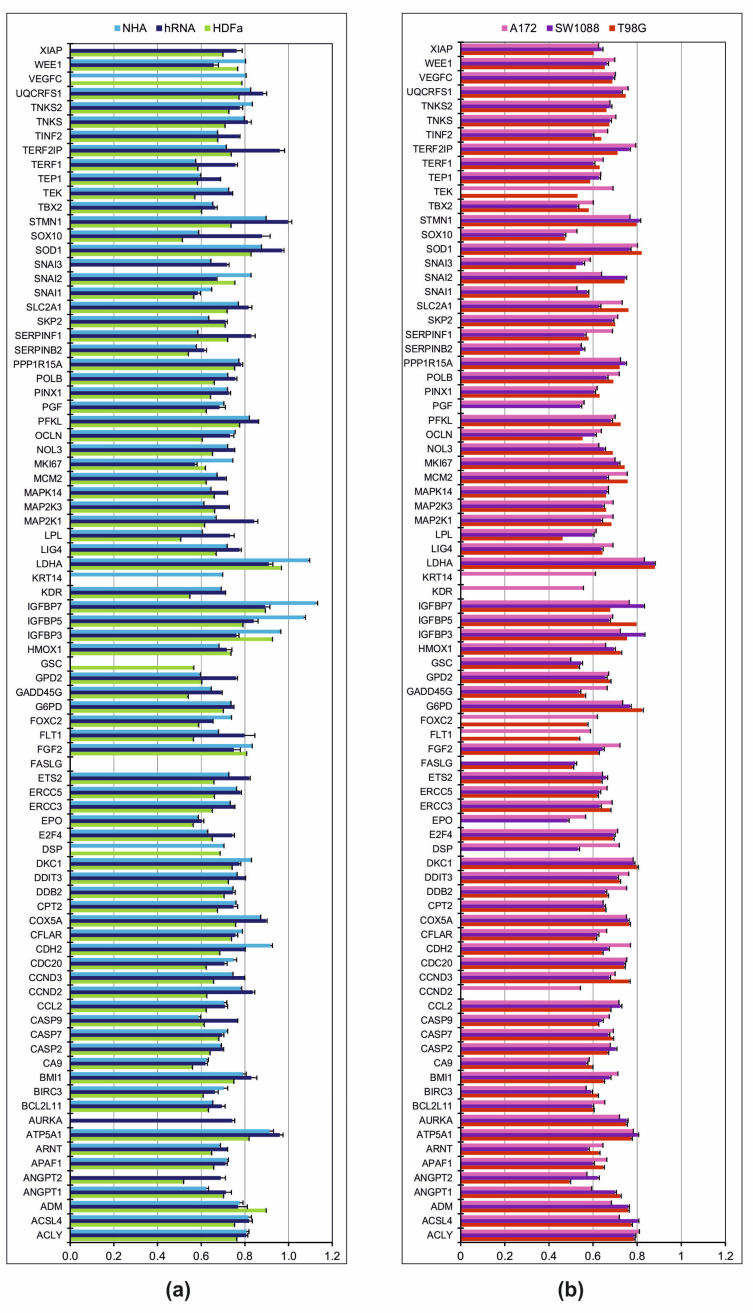
The mean of Ct ratio of mRNA levels. (**a**) The mRNA expression levels in control group. (**b**) The mRNA expression levels in group of tumour cell lines. Ct ratio, gene of interest/housekeeping genes. Each mRNA level included three replicates.

**Figure 2 ijms-23-10883-f002:**
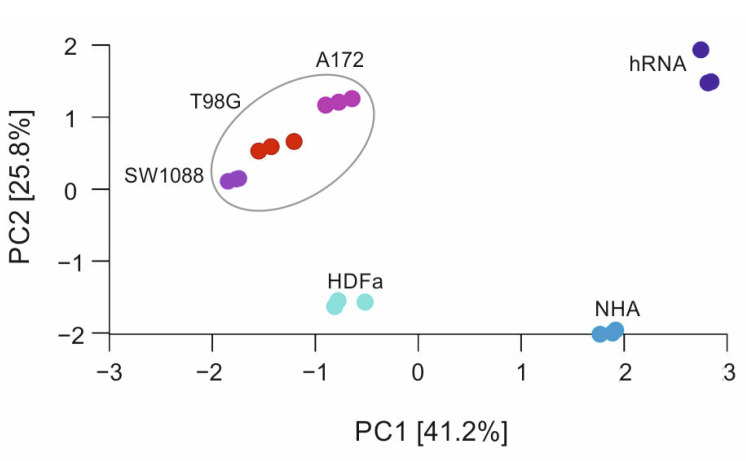
Scatter plot of principal component analysis. The grey elliptical line represents data of tumour cell lines, dots outside the line are control samples. All values of each sample were measured in triplicates.

**Figure 3 ijms-23-10883-f003:**
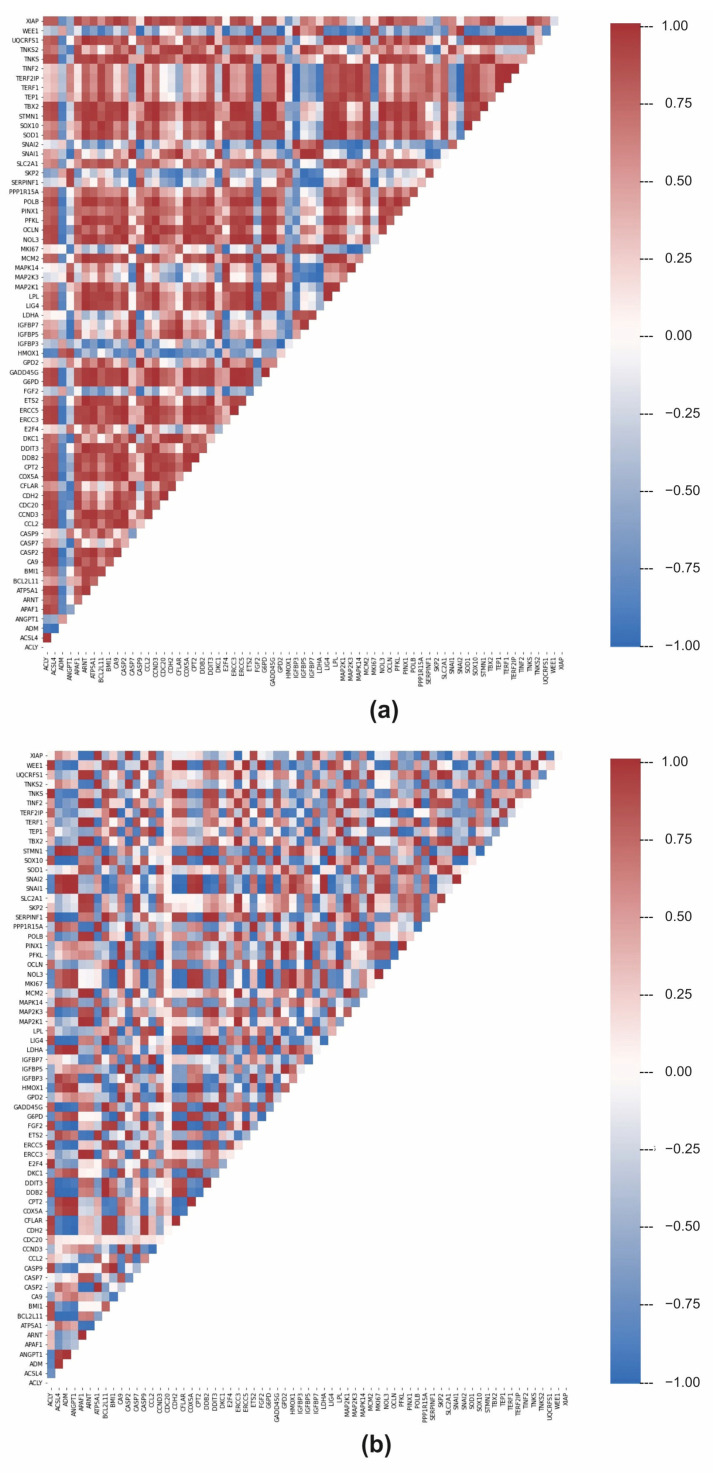
Pearson correlation of individual genes expressions of control (**a**) and tumour (**b**) cell lines.

**Figure 4 ijms-23-10883-f004:**
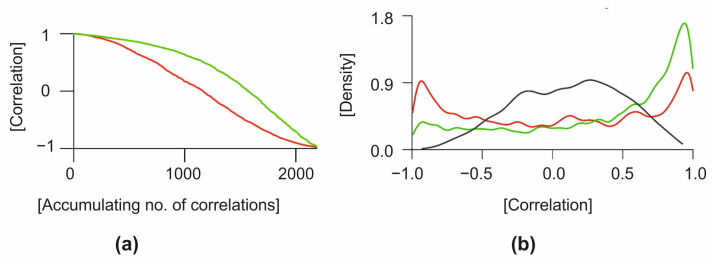
Statistical analysis of gene expression correlations. (**a**) Descending order of correlations. (**b**) Kernel density of correlations. The green line represents data from control, red line from tumour and grey line is the mix of both groups.

**Figure 5 ijms-23-10883-f005:**
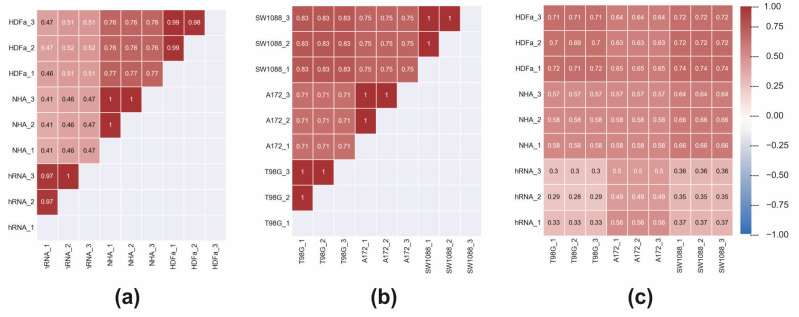
Pearson correlations between gene expression profiles of analysed cell lines. (**a**) Correlation of gene expression profiles within control group, (**b**) within tumour group and (**c**) between samples of control and tumour group.

**Figure 6 ijms-23-10883-f006:**
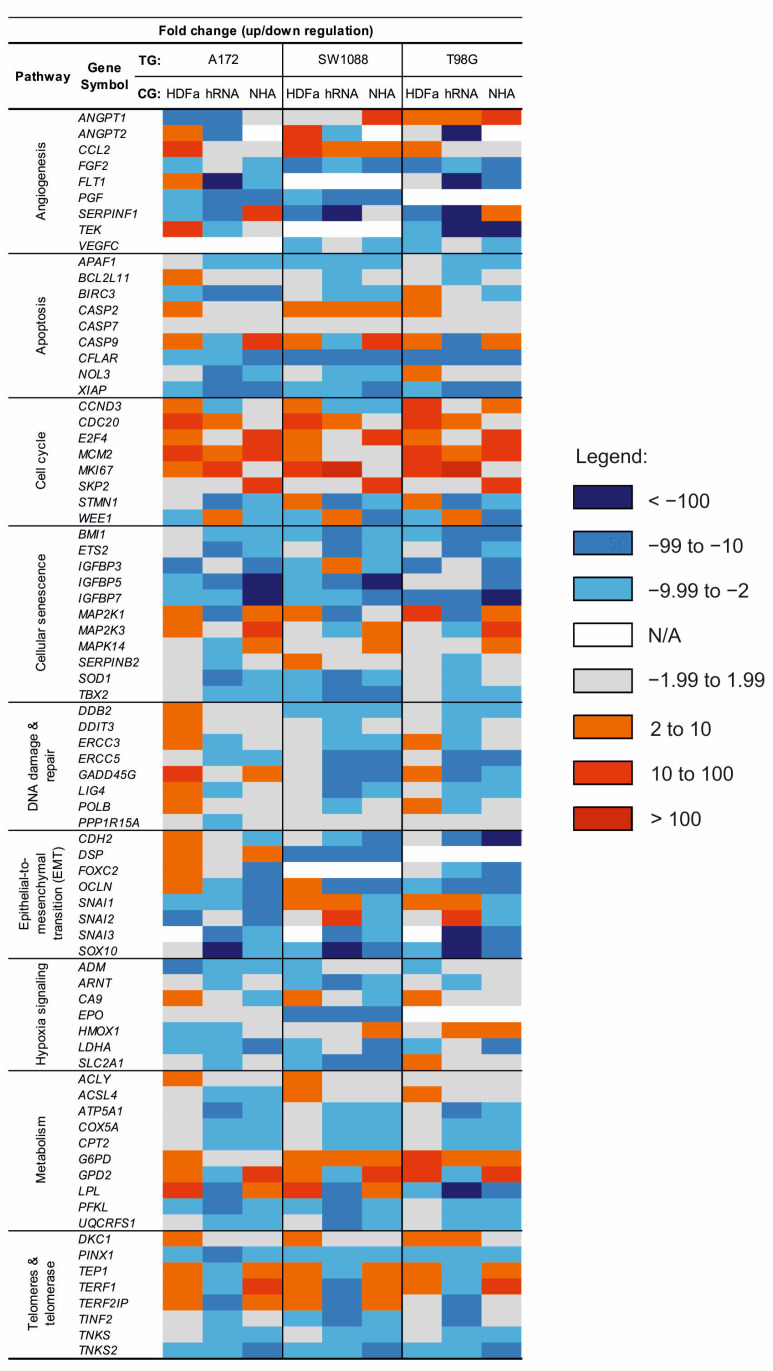
Heat map of fold change regulation between test and control group. Test group (TG): A172, SW1088 and T98G; Control group (CG): HDFa, hRNA and NHA.

**Figure 7 ijms-23-10883-f007:**
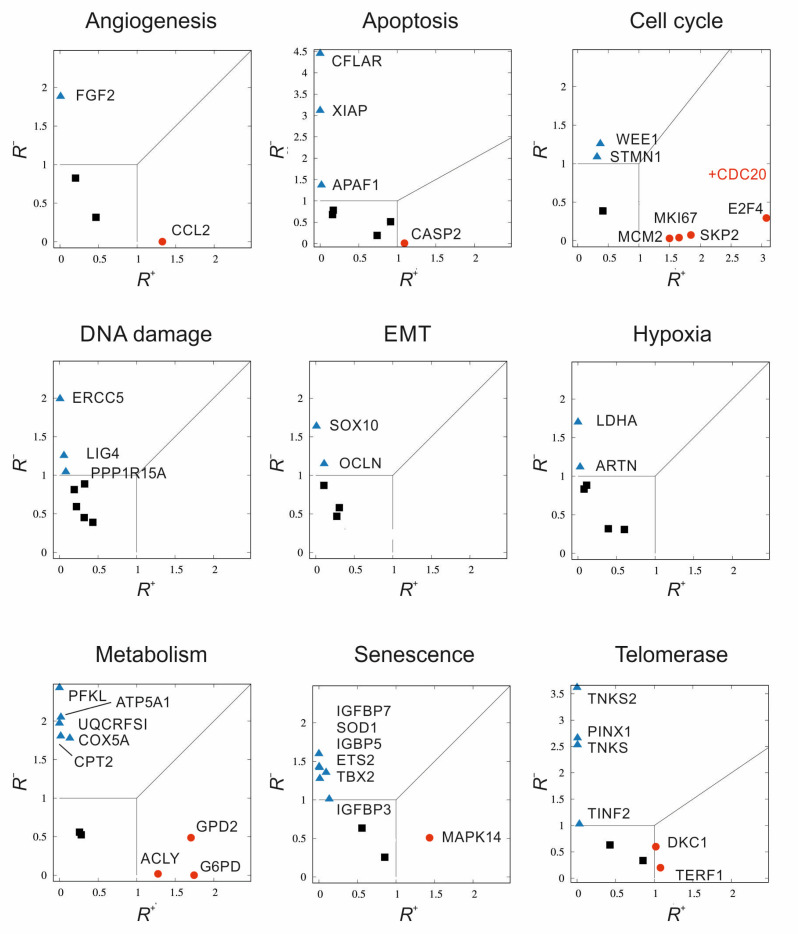
Multi-criterial correlation plots of gene expression in tumour cell lines relative to the controls. *R*^+^ represents a negative proportion of the numerator’s dissimilarity tendencies, *R*^−^ positive. A 45-degree line separates the regions of overexpression and underexpression; at this line, overexpression and underexpression are therefore balanced. The scenarios when noise predominates are covered by the (0.1) × (0.1) square. There are three main regions in the graph to categorize according to the level of gene expression with three ways of marking the corresponding points: statistically significant and overexpressed with *R*^+^ > 1 (red circles), statistically significant underexpressed with *R*^−^ > 1 (blue triangles), finally statistically less significant (black squares) bordered by 0 ≤ *R*^+^ ≤ 1, 0 ≤ *R*^−^ ≤ 1.

**Table 1 ijms-23-10883-t001:** List of analysed genes and their assignment to cellular pathways.

Pathway	Symbol	Gene Name
Angiogenesis	*ANGPT1*	Angiopoietin 1
	*ANGPT2*	Angiopoietin 2
	*CCL2*	Chemokine (C-C motif) ligand 2
	*FGF2*	Fibroblast growth factor 2
	*FLT1*	Fms-related tyrosine kinase 1
	*KDR*	Kinase insert domain receptor
	*PGF*	Placental growth factor
	*SERPINF1*	Serpin peptidase inhibitor, clade F member 1
	*TEK*	TEK tyrosine kinase, endothelial
	*VEGFC*	Vascular endothelial growth factor C
Apoptosis	*APAF1*	Apoptotic peptidase activating factor 1
	*BCL2L11*	BCL2-like 11 (apoptosis facilitator)
	*BIRC3*	Baculoviral IAP repeat containing 3
	*CASP2*	Caspase 2
	*CASP7*	Caspase 7
	*CASP9*	Caspase 9
	*CFLAR*	CASP8 and FADD-like apoptosis regulator
	*FASLG*	Fas ligand (TNF superfamily, member 6)
	*NOL3*	Nucleolar protein 3 (apoptosis repressor)
	*XIAP*	X-linked inhibitor of apoptosis
Cell cycle	*AURKA*	Aurora kinase A
	*CCND2*	Cyclin D2
	*CCND3*	Cyclin D3
	*CDC20*	Cell division cycle 20 homolog (S. cerevisiae)
	*E2F4*	E2F transcription factor 4, p107/p130-binding
	*MCM2*	Minichromosome maintenance complex component 2
	*MKI67*	Antigen identified by monoclonal antibody Ki-67
	*SKP2*	S-phase kinase-associated protein 2 (p45)
	*STMN1*	Stathmin 1
	*WEE1*	WEE1 homolog (S. pombe)
Cellular senescence	*BMI1*	BMI1 polycomb ring finger oncogene
	*ETS2*	V-Ets erythroblastosis virus E26 oncogene homolog 2
	*IGFBP3*	Insulin-like growth factor binding protein 3
	*IGFBP5*	Insulin-like growth factor binding protein 5
	*IGFBP7*	Insulin-like growth factor binding protein 7
	*MAP2K1*	Mitogen-activated protein kinase kinase 1
	*MAP2K3*	Mitogen-activated protein kinase kinase 3
	*MAPK14*	Mitogen-activated protein kinase 14
	*SERPINB2*	Serpin peptidase inhibitor, clade B, member 2
	*SOD1*	Superoxide dismutase 1, soluble
	*TBX2*	T-box 2
DNA damage and repair	*DDB2*	Damage-specific DNA binding protein 2, 48kDa
	*DDIT3*	DNA-damage-inducible transcript 3
	*ERCC3*	Excision repair cross-complementing rodent repair deficiency, complementation group 3
	*ERCC5*	Excision repair cross-complementing rodent repair deficiency, complementation group 5
	*GADD45G*	Growth arrest and DNA-damage-inducible, γ
	*LIG4*	DNA Ligase 4, ATP-dependent
	*POLB*	DNA Polymerase beta
	*PPP1R15A*	Protein phosphatase 1, regulatory subunit 15A
Epithelial-to-mesenchymal transition (EMT)	*CDH2*	Cadherin 2, type 1, N-cadherin (neuronal)
	*DSP*	Desmoplakin
	*FOXC2*	Forkhead box C2
	*GSC*	Goosecoid homeobox
	*KRT14*	Keratin 14
	*OCLN*	Occludin
	*SNAI1*	Snail homolog 1 (Drosophila)
	*SNAI2*	Snail homolog 2 (Drosophila)
	*SNAI3*	Snail homolog 3 (Drosophila)
	*SOX10*	SRY (sex determining region Y)-box 10
Hypoxiasignalling	*ADM*	Adrenomedullin
	*ARNT*	Aryl hydrocarbon receptor nuclear translocator
	*CA9*	Carbonic anhydrase 9
	*EPO*	Erythropoietin
	*HMOX1*	Heme oxygenase 1
	*LDHA*	Lactate dehydrogenase A
	*SLC2A1*	Solute carrier family 2, member 1
Metabolism	*ACLY*	ATP citrate lyase
	*ACSL4*	Acyl-CoA synthetase long-chain family member 4
	*ATP5A1*	Mitochondrial ATP synthase alpha subunit 1
	*COX5A*	Cytochrome c oxidase subunit 5A
	*CPT2*	Carnitine palmitoyltransferase 2
	*G6PD*	Glucose-6-phosphate dehydrogenase
	*GPD2*	Glycerol-3-phosphate dehydrogenase 2 (mitochondrial)
	*LPL*	Lipoprotein lipase
	*PFKL*	Phosphofructokinase, liver
	*UQCRFS1*	Ubiquinol-cytochrome c reductase Rieske iron-sulfur polypeptide 1
Telomeres and telomerase	*DKC1*	Dyskerin
	*PINX1*	PIN2/TERF1 interacting, telomerase inhibitor 1
	*TEP1*	Telomerase-associated protein 1
	*TERF1*	Telomeric repeat binding factor (NIMA-interacting) 1
	*TERF2IP*	Telomeric repeat binding factor 2, interacting protein
	*TINF2*	TERF1-interacting nuclear factor 2
	*TNKS*	Tankyrase
	*TNKS2*	Tankyrase 2

## Data Availability

The data generated and analysed during the current study are available from the corresponding author upon reasonable request.

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
