# Peer review of "Different Approaches for the Profiling of Cancer Pathway-Related Genes in Glioblastoma Cells"

_ijms, 2022, doi:10.3390/ijms231810883_

Round 1
Reviewer 1 Report
An interesting study comparing the expression of a variety of cancer-relevant genes in human brain cancer cell lines and some normal controls.
There are several ways that this manuscript could be improved.
Abstract
Line 25/26: “Statistical adjustments, such as PCA or LDA, were used to assess absolute gene levels and fold regulatory changes, respectively.”
I’m not certain that this statement is true. PCA and LDA (and PCA-LDA as in this case) do not directly assay the absolute gene expression levels or fold changes and in this paper no specific genes are identified using these methods.
Results:
It would be helpful if a summary was given near the start of the results section of i.) the exact number of genes assayed by the Human Cancer PathwayFinder PCR Array is specified, i.) the number of genes that were rejected based on quality control (QC) metrics e.g. genes that produced no results. There are several numbers throughout the paper and it is difficult to keep track of them.
Figure 1: this could be improved in several ways; i.) It would be easier to read if in color. As there are six categories (3 cell lines and 3 controls), you could use a simple color scheme such as red, green, blue for one group and cyan, magenta, yellow for the other. ii.) remove the title of the plot and put the “GOI” and “HKG” (not as acronyms, but full words) at the top of each column. iii.) this is the only plot that includes all genes in reverse alphabetical order. It would be more clear if the genes were in the same order as Table 1 and labels were added to show the different pathway groups.
Section 2.3.1:
Lines 137 to 139. I disagree that the overlap between HDFa and A172 is interesting; one could equally point out the overlap in PC2 between A172 and hRNA. I don’t think these overlaps are hugely meaningful.
Figure 2: this figure would be improved by using the same color scheme proposed for Figure 1 and potentially adding a legend to the figure, although the text labels used are very useful.
Section 2.3.2:
Lines 148-158. I disagree with the conclusions. I'm not sure that PCA-LDA supports the idea that specific genes are driving the differences between the groups; in this analysis, PCA and LDA are not used to pick out specific genes. It is clear from the PCA that the tumors and controls cluster separately, so I am not sure why the LDA is required to further illustrate this point.
Section 2.3.3:
Line 169; would be better if the word “eliminated” was changed to “dysregulated”. Eliminated is a very strong word and cancer cells are not completely chaotic, or they would not survive.
Figure 4: Shouldn’t correlation analysis be done for each sample individually, not by combining them all into two groups? I think the aggregate analysis is useful and shows an interesting feature, but the argument would be stronger if each sample were studied independently, even if this way put in the supplementary information (SI).
Speaking of the SI, Figure S1 is not in the SI. Please include it. It would also be helpful to include the raw data in the SI. This would allow other researchers to replicate your results and/or use them in their own studies.
Section 2.4:
Figure 5: was any clustering attempted? This could potentially reveal relationships between the samples and the genes, and should probably be included in the SI.
All of the text in section 2.4.1 to 2.4.9 should be simplified and condensed. I think there is danger of over-analyzing what is a very small dataset and looking too deeply into these results. Although I have used few words here, I think that this is my single largest issue with the manuscript.
Section 2.5:
Figure 7: this is a good figure but needs some improvement. It would be very helpful to add either a legend to the plots or more explanation in the text legend. I can see that blue triangles are genes with R- above 1, red circles have R+ above 1 and black squares are genes that have neither, but this isn’t explained. This might be the most interesting figure but it needs a little more explanation.
Perhaps combing sections 2.4 and 2.5 would be beneficial and simplify the message of the manuscript?
Section 3:
Line 375: it would be very helpful if the origins of the cell lines were made more clear at the beginning of the paper.
The discussion is well written, but is too long and too detailed given the small size of the data set (similar to my comments about section 2.4). This would benefit from being simplified and condensed, there is too much information from too little data.
Reviewer 2 Report
The authors described the evaluation of gene expression analysis on astrocyte-derived cancer and three normal control by qRT-PCR. Although their approach is attractive, further clarification is needed to fully support this review.
Major revision
1. Recently, microarray or RNA-Seq analysis are popular methods of gene expression analysis. Please provide explanations of comparable considerations of the authors' evaluation method and list its disadvantages and advantages.
2. Please justify that protein expression was not evaluated.
3. Tittle is not suitable for this study.
4. “Cancer pathway” is not precise in this manuscript including title, abstract, and body. Isn't the correct term "Cancer-related pathway"?
Minor revision
Line 25: Please spell out PCA and LDA.
Line 27: 72 cases? Isn’t this “genes”?
Line 27: Please spell out “Ct”.
line 44: Put reference 2 with reference 3 at the end of the sentence.
Line 50-52: Please add the appropriate references.
Line 55: Please formally describe p53. isn't it “53-kilodalton protein”?
Line 58: Correctly, “gene levels” is “gene expression levels”.
Round 2
Reviewer 2 Report
The authors adequately answer my concerns.